# Discovery from *Hypericum elatoides* and synthesis of hyperelanitriles as α-aminopropionitrile-containing polycyclic polyprenylated acylphloroglucinols

Jin-Yan Xie[1], Pengfei Li[1], Xi-Tao Yan [1✉] & Jin-Ming Gao [1✉]

The search for lead compounds with anti-neuroinflammatory activity from structurally 'optimized' natural products is a crucial and promising strategy in the quest to discover safe and efficacious agents for treating neurodegenerative diseases. A phytochemical investigation on the aerial portions of *Hypericum elatoides* led to the isolation of five nitrogenous polycyclic polyprenylated acylphloroglucinols (PPAPs), hyperelanitriles A–D (**1**–**4**) and hyperelamine A (**5**). Their structures were determined by spectroscopic analysis, ECD and NMR calculations, and X-ray crystallography. To the best of our knowledge, compounds **1**–**4** represent the first examples of acylphloroglucinols featuring an α-aminonitrile moiety, while **5** is a rare enamine-containing PPAP. Further, the synthesis of these naturally occurring PPAP-based nitriles or amines was accomplished. Compound **5** exhibited inhibitory activity against LPS-activated NO production in BV-2 cells, potentially through the suppression of TLR-4/NF-κB signaling. Here we show the isolation, structural elucidation, synthesis, and bioactive evaluation of compounds **1**–**5**.

---

[1] Shaanxi Key Laboratory of Natural Products & Chemical Biology, College of Chemistry & Pharmacy, Northwest A&F University, 712100 Yangling, People's Republic of China. ✉email: xty@nwsuaf.edu.cn; jinminggao@nwsuaf.edu.cn

In recent years, polycyclic polyprenylated acylphloroglucinols (PPAPs) have attracted considerable attention from chemists and pharmacologists due to their diverse and complex carbon skeletons and significant bioactivity[1–5]. PPAPs found mainly in plants of the genera *Hypericum* and *Garcinia* are a special class of fascinating natural products composed of a highly oxygenated acylphloroglucinol-derived cores and dense prenyl or geranyl side chains[1,6]. According to the different structural features, the various PPAPs are typically divided into three groups: the bicyclic polyprenylated acylphloroglucinols (BPAPs) with major bicyclo[3.3.1] nonane-2,4,9-trione core, the caged PPAPs with adamantane and homoadamantane scaffolds, spirocyclic PPAPs, and other complicated PPAPs[1,7–9]. Up to date, more than 1100 PPAPs have been characterized[10,11]. However, the naturally occurring PPAPs possessing the nitrile or amine group have not been found.

*Hypericum elatoides*, a member of the genus *Hypericum* (Hypericaceae), is a perennial herb mainly distributed in Shaanxi and Gansu Provinces, northwestern China. In our ongoing search for novel and bioactive natural products from this plant[12–14], five uncommon PPAPs, hyperelanitriles A–D (1–4) and hyperelamine A (5), were isolated from the aerial parts of *H. elatoides*. Their structures, including absolute configurations, were elucidated based on a combination of extensive spectroscopic data analysis, $^{13}$C NMR calculations coupled with DP4+ probability analysis, ECD calculations, single-crystal X-ray diffraction, and chemical synthesis. Structurally, compounds 1–4 are the first examples of acylphloroglucinols possessing an α-aminopropionitrile moiety, while compound 5 contains an unusual enamine group based on a BPAP architecture. Herein, we describe the structural elucidation, synthesis, and anti-neuroinflammatory evaluation of these nitrogenous PPAPs as well as the underlying mechanism of compound 5.

## Results and discussion

**Structural elucidation.** The powdered aerial portions of *H. elatoides* were extracted using MeOH and then partitioned with petroleum ether. The petroleum ether-soluble part was fractionated by silica gel column chromatography (CC) and further purified by repeated CC over reversed-phase $C_{18}$, and finally semipreparative HPLC to afford compounds 1–5 (Fig. 1).

Hyperelanitrile A (1) presented a molecular formula of $C_{37}H_{48}N_2O_3$ with 15 degrees of unsaturation, based on the HRESIMS ion peak at $m/z$ 591.3559 $[M + Na]^+$ (calcd. for 591.3557). The IR spectrum showed absorptions at 2308, 1722, and 1097 $cm^{-1}$, corresponding to the nitrile, carbonyl, and amine groups, respectively. The $^1$H NMR spectrum of 1, combined with the analysis of the HSQC spectrum, revealed the presence of a NH group with a dramatic downfield proton [$\delta_H$ 13.33 (d, $J = 8.7$ Hz)] due to strong intramolecular hydrogen bonding, one unusual monosubstituted benzene moiety with five nonequivalent protons [$\delta_H$ 6.99 (d, $J = 7.6$ Hz), 7.44 (t, $J = 7.6$ Hz), 7.50 (t, $J = 7.6$ Hz), 7.54 (t, $J = 7.6$ Hz), and 7.19 (d, $J = 7.6$ Hz)], three trisubstituted olefins ($\delta_H$ 5.01, 4.90, and 4.81), two sp$^3$ methines ($\delta_H$ 4.16 and 1.71), five sp$^3$ methylenes ($\delta_H$ 2.46, 2.40, 2.22, 2.18, 1.98, 1.93, 1.82, 1.77, 1.32,

1.15), and nine methyls [$\delta_H$ 1.70 (s), 1.66 (s), 1.65 (s), 1.63 (d, $J = 7.1$ Hz), 1.55 (s), 1.51 (s), 1.40 (s), 1.35 (s), and 1.10 (s)] (Table 1). The $^{13}$C NMR spectrum along with DEPT spectra exhibited 37 carbon signals, including characteristic resonances for a bicyclo[3.3.1]nonane-2,4,9-trione system with three nonconjugated carbonyl groups ($\delta_C$ 210.2, 199.8, and 194.3) and three nonoxygenated quaternary carbons ($\delta_C$ 69.4, 59.4, and 51.3), three prenyl groups (including three double bonds, three methylenes, and six vinylic methyls), and one phenyl group. These analyses suggested that 1 was a type B PPAP derivative. Compared to hyperelatone A (1a), which is the first PPAP isolated from *H. elatoides*[13], the 1D NMR data of 1 displayed an upfield-shifted chemical shift for C-15 ($\delta_C$ 170.0 in 1 vs. $\delta_C$ 197.0 in 1a) and three additional signals comprising a quaternary carbon at $\delta_C$ 117.4, a methine at $\delta_C$ 40.1, and a methyl at $\delta_C$ 19.9. Considering the presence of an NH group and one more nitrogen atom required by the molecular formula and the remaining three degrees of unsaturation, these differences were attributable to the attachment of an α-aminopropionitrile fragment to C-15 and the formation of an enamine group at the C-3–C-15 double bond. This assignment was supported by the $^1$H–$^1$H COSY correlations of H$_3$-24/H-22/NH as well as the HMBC correlations from H$_3$-24 to C-23 and C-22, from H-22 to C-15, C-23, and C-24, and from the NH proton to C-3, C-16, and C-22 (Fig. 2). The remainder of the structure of 1, similar to 1a, was established by extensive analysis of its 2D NMR spectra. Accordingly, the planar structure of 1 was established as a type B PPAP with an α-aminopropionitrile side chain linked to C-15. It is worth noting that the protons and carbons on opposite sides of this monosubstituted phenyl ring are not chemically equivalent even though there is a plane of symmetry. This may result from the α-aminopropionitrile substituent at C-15 increased the steric hindrance around the Ar–C-15 axis and raised the inversion barrier.

The relative configuration of 1 was determined by analysis of ROESY correlations (Fig. 3). The correlations of H-6a/H-27, H-27/H-26b, and H-26b/H$_3$-37 indicated that these protons were cofacial and assigned randomly as β-oriented, while the cross-peaks of H-6b/H-7, H-6b/H$_3$-25, H-6b/H-31b, H-7/H-32a, H-10a/H-31a, and H-10b/H-31a indicated the α-orientation of these protons. Furthermore, key ROESY correlations of H-17/H-11, H-17/H$_3$-13, and H-18/H$_3$-13 revealed an *E*-configuration for the double bond between C-3 and C-15. However, the configuration of C-22 remained unassigned due to the absence of reliable ROESY correlations. Subsequently, an X-ray diffraction experiment (Fig. 4) under Cu Kα radiation [Flack parameter = 0.14(7), CCDC 2151888] was successfully performed, suggesting a 22*R* configuration for 1. To further validate the absolute configuration, quantum chemical $^{13}$C NMR calculations of three isomers [(1*S*,3*E*,5*R*,7*S*,8*R*,22*R*)-1, (1*S*,3*Z*,5*R*,7*S*,8*R*,22*R*)-1, and (1*S*,3*E*,5*R*,7*S*,8*R*,22*S*)-1] together with the experimental values of 1 were applied to the DP4+ analysis. The results confirmed that the (1*S*,3*E*,5*R*,7*S*,8*R*,22*R*)-1 was the most reliable configuration with 99.99% probability (Supplementary Table S2 and Figs. S2 and S3). Thus, the absolute configuration of 1 was determined to be 1*S*, 5*R*, 7*S*, 8*R*, and 22*R*.

Hyperelanitrile B (2) was obtained as an inseparable mixture with 1 when in solution. Nevertheless, its structure could be characterized unambiguously because it showed distinctive NMR signals. On the basis of 1D and 2D NMR analysis, the planar structure of 2 was determined to be identical to that of 1. Their notable differences were found to be the chemical shifts of C-2 and C-4 (Table 1). The carbonyl C-2 resonance shifted downfield significantly from $\delta_C$ 194.3 in 1 to $\delta_C$ 199.8 in 2, while the carbonyl C-4 signal shifted upfield significantly from $\delta_C$ 199.8 in 1 to $\delta_C$ 193.8 in 2. These observations indicated the intramolecular hydrogen bonding between the NH proton and the carbonyl group at C-4 in 1 changed to the carbonyl group at C-2 in 2,

**Fig. 1 Chemical structures.** Structures of hyperelanitriles A–D (1–4) and hyperelamine A (5) from *Hypericum elatoides*.

**Table 1 ¹H (800 MHz) and ¹³C NMR (200 MHz) spectroscopic data for compounds 1–4 in CDCl₃ ($\delta$ in ppm, $J$ in Hz).**

| No. | 1 $\delta_H$ | 1 $\delta_C$ | 2 $\delta_H$ | 2 $\delta_C$ | 3 $\delta_H$ | 3 $\delta_C$ | 4 $\delta_H$ | 4 $\delta_C$ |
|---|---|---|---|---|---|---|---|---|
| 1 | | 69.4, C | | 70.0, C | | 69.3, C | | 70.1, C |
| 2 | | 194.3, C | | 199.8, C | | 194.1, C | | 200.3, C |
| 3 | | 112.7, C | | 112.6, C | | 112.0, C | | 113.0, C |
| 4 | | 199.8, C | | 193.8, C | | 199.5, C | | 193.8, C |
| 5 | | 59.4, C | | 59.8, C | | 59.4, C | | 59.9, C |
| 6a | 2.22, br d (14.3) | 41.4, CH₂ | 2.07, dd (14.3, 1.6) | 41.7, CH₂ | 2.25, br d (14.2) | 41.3, CH₂ | 2.06, dd (14.2, 1.8) | 41.8, CH₂ |
| 6b | 1.93, dd (14.3, 7.1) | | 1.80, dd (14.3, 6.8) | | 1.92, dd (14.2, 7.2) | | 1.81, dd (14.2, 6.6) | |
| 7 | 1.71, m | 39.7, CH | 1.71, m | 40.0, CH | 1.72, m | 39.6, CH | 1.72, m | 40.1, CH |
| 8 | | 51.3, C | | 51.4, C | | 51.3, C | | 51.4, C |
| 9 | | 210.2, C | | 210.8, C | | 210.6, C | | 210.5, C |
| 10a | 2.46, dd (13.5, 5.4) | 26.2, CH₂ | 2.66, dd (13.5, 8.5) | 26.3, CH₂ | 2.46, dd (13.5, 5.2) | 26.2, CH₂ | 2.68, dd (13.4, 8.3) | 26.4, CH₂ |
| 10b | 2.40, dd (13.5, 8.5) | | 2.64, dd (13.5, 5.8) | | 2.42, dd (13.5, 8.5) | | 2.64, dd (13.4, 5.0) | |
| 11 | 4.81, t (6.2) | 120.0, CH | 4.81, t (6.2) | 120.3, CH | 4.84, t (6.8) | 120.3, CH | 4.87, t (6.3) | 119.5, CH |
| 12 | | 133.6, C | | 133.4, C | | 133.4, C | | 134.4, C |
| 13 | 1.66, s | 26.1, CH₃ | 1.57, s | 26.1, CH₃ | 1.67, s | 26.1, CH₃ | 1.65, s | 25.9, CH₃ |
| 14 | 1.40, s | 17.8, CH₃ | 1.70, s | 18.2, CH₃ | 1.43, s | 17.9, CH₃ | 1.73, s | 18.2, CH₃ |
| 15 | | 170.0, C | | 169.2, C | | 170.2, C | | 169.1, C |
| 16 | | 132.3, C | | 132.5, C | | 132.5, C | | 132.4, C |
| 17 | 6.99, d (7.6) | 125.8, CH | 7.40, d (7.6) | 126.8, CH | 6.85, d (7.6) | 125.5, CH | 7.14, d (7.6) | 125.8, CH |
| 18 | 7.44, t (7.6) | 129.3, CH | 7.57, t (7.6) | 129.3, CH | 7.38, t (7.6) | 128.9, CH | 7.51, overlap | 128.8, CH |
| 19 | 7.50, t (7.6) | 129.7, CH | 7.51, t (7.6) | 130.5, CH | 7.51, t (7.6) | 129.9, CH | 7.50, overlap | 130.2, CH |
| 20 | 7.54, t (7.6) | 128.8, CH | 7.40, t (7.6) | 129.4, CH | 7.60, t (7.6) | 129.2, CH | 7.46, t (7.6) | 129.8, CH |
| 21 | 7.19, d (7.6) | 126.1, CH | 6.88, d (7.6) | 126.4, CH | 7.45, d (7.6) | 127.0, CH | 7.03, d (7.6) | 126.5, CH |
| 22 | 4.16, dq (8.7, 7.1) | 40.1, CH | 4.11, dq (9.0, 7.0) | 41.1, CH | 4.04, dq (8.8, 7.1) | 41.0, CH | 4.19, dq (8.8, 7.3) | 40.2, CH |
| 23 | | 117.4, C | | 117.9, C | | 117.8, C | | 117.3, C |
| 24 | 1.63, d (7.1) | 19.9, CH₃ | 1.48, d (7.0) | 20.6, CH₃ | 1.54, d (7.1) | 20.4, CH₃ | 1.68, d (7.3) | 20.0, CH₃ |
| 25 | 1.35, s | 18.5, CH₃ | 1.08, s | 18.1, CH₃ | 1.32, s | 18.5, CH₃ | 1.08, s | 18.0, CH₃ |
| 26a | 2.18, m | 28.7, CH₂ | 2.12, m | 28.9, CH₂ | 2.23, m | 28.7, CH₂ | 2.11, m | 28.9, CH₂ |
| 26b | 1.98, m | | 2.03, m | | 1.97, m | | 2.01, m | |
| 27 | 4.90, t (6.9) | 124.9, CH | 4.88, t (6.8) | 124.1, CH | 4.88, t (6.9) | 124.5, CH | 4.90, t (6.8) | 124.3, CH |
| 28 | | 132.3, C | | 132.8, C | | 132.8, C | | 132.6, C |
| 29 | 1.70, s | 25.8, CH₃ | 1.64, s | 25.7, CH₃ | 1.71, s | 25.8, CH₃ | 1.65, s | 25.7, CH₃ |
| 30 | 1.51, s | 17.7, CH₃ | 1.51, s | 17.9, CH₃ | 1.58, s | 17.8, CH₃ | 1.49, s | 17.9, CH₃ |
| 31a | 1.32, m | 35.8, CH₂ | 1.41, m | 36.1, CH₂ | 1.32, m | 35.9, CH₂ | 1.41, m | 36.1, CH₂ |
| 31b | 1.15, m | | 1.21, m | | 1.14, m | | 1.21, m | |
| 32a | 1.82, m | 22.4, CH₂ | 1.89, m | 22.6, CH₂ | 1.83, m | 22.4, CH₂ | 1.89, m | 22.6, CH₂ |
| 32b | 1.77, m | | 1.82, m | | 1.78, m | | 1.83, m | |
| 33 | 5.01, t (7.0) | 123.9, CH | 5.03, t (7.0) | 123.9, CH | 5.00, t (7.0) | 123.9, CH | 5.03, t (7.0) | 123.9, CH |
| 34 | | 131.9, C | | 131.9, C | | 131.8, C | | 131.9, C |
| 35 | 1.65, s | 25.7, CH₃ | 1.66, s | 25.7, CH₃ | 1.65, s | 25.7, CH₃ | 1.66, s | 25.7, CH₃ |
| 36 | 1.55, s | 17.6, CH₃ | 1.57, s | 17.6, CH₃ | 1.55, s | 17.6, CH₃ | 1.57, s | 17.6, CH₃ |
| 37 | 1.10, s | 19.1, CH₃ | 1.31, s | 19.2, CH₃ | 1.10, s | 19.1, CH₃ | 1.30, s | 19.2, CH₃ |
| –NH | 13.33, d (8.7) | | 13.20, d (9.0) | | 13.49, d (8.8) | | 13.06, d (8.8) | |

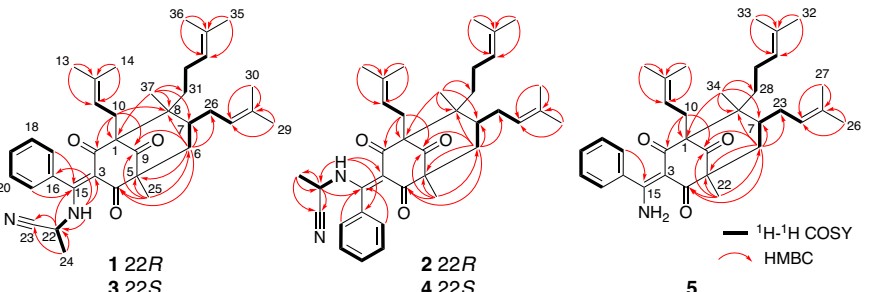

**Fig. 2 ¹H-¹H COSY and HMBC correlations for compounds 1–5.** Black bold bonds represent COSY correlations. Red arrows represent key HMBC correlations.

which was likely caused due to a *Z*-configuration of the double bond between C-3 and C-15 in **2**. This assumption was further supported by the absence of any ROESY signal between the protons of the benzene ring and the prenyl group at C-1. Moreover, the remaining ROESY correlations of **2** were identical to those of **1** (Fig. 3), indicating that **2** had the same relative configuration as **1** except for the 3*Z*-configured double bond. Finally, DP4+ calculations (Supplementary Fig. S6) provided a 100.00% probability of (1*S*,3*Z*,5*R*,7*S*,8*R*,22*R*)-**2** being the correct stereoisomer and confirmed the 3*Z* double bond geometry of **2**.

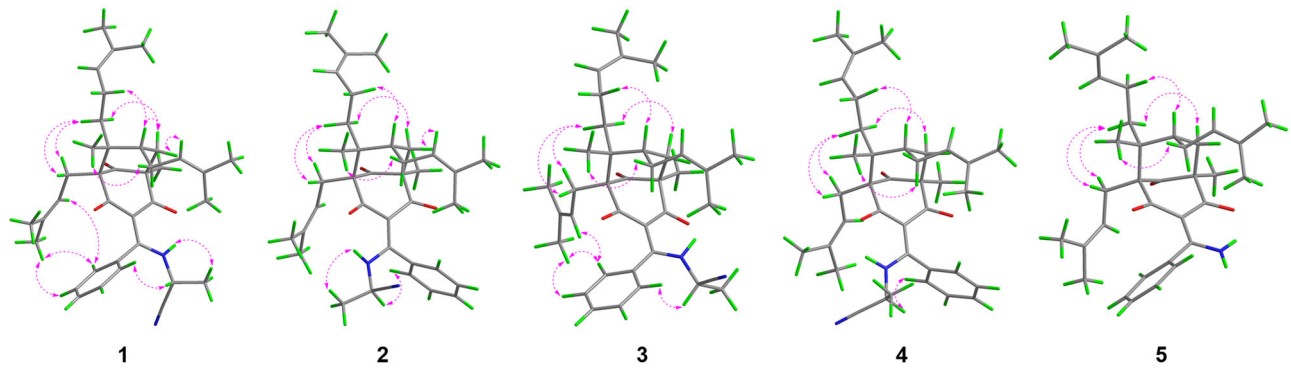

**Fig. 3 Key ROESY correlations of compounds 1–5.** Pink dashed arrows represent selected ROESY correlations.

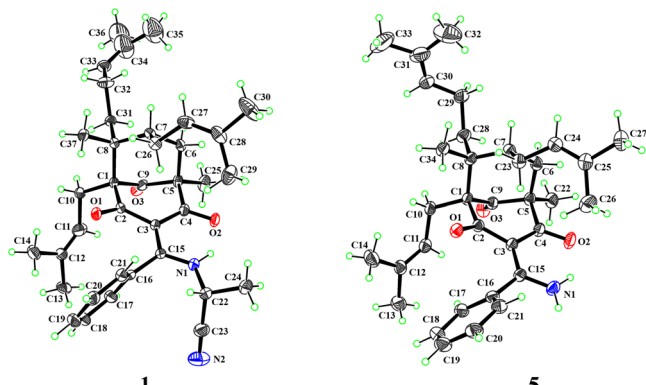

**Fig. 4 X-ray crystallographic structures.** ORTEP drawings of compounds **1** and **5**. Displacement ellipsoids are drawn at the 30% probability level.

Hyperelanitrile C (**3**) possessed the same molecular formula ($C_{37}H_{48}N_2O_3$) as **1**, inferred from a sodium adduct ion peak at $m/z$ 591.3558 in the HRESIMS. The extensive analysis of 1D and 2D NMR data indicated that **3** shared the same planar structure as **1**. The relative configurations of C-1, C-5, C-7, and C-8 of **3** were assigned to be the same as those of **1** based on their nearly identical ROESY correlations (Fig. 3). Additionally, the 3*E* geometry of the double bond was established by the crucial ROESY correlations of H-17/H₃-13, H-17/H-11, and H-18/H₃-13. These findings suggest that **3** differs from **1** only in the chiral center at C-22, and is a C-22 epimer of **1**. ¹³C NMR calculations followed by DP4+ analysis (Supplementary Fig. S9) indicated 22*S* as correct configuration with 99.61% probability between the possible diastereomers (22*R* and 22*S*). Finally, the absolute configuration of **3** was confirmed as 1*S*, 5*R*, 7*S*, 8*R*, and 22*S* by comparison of the calculated and experimental ECD curves (Supplementary Fig. S10).

Hyperelanitrile D (**4**) was shown to be an inseparable mixture with **3**. A careful comparison of the NMR data of **4** with those of **3** revealed that the only difference was a *Z*-configuration in **4** instead of an *E*-configuration in **3** for the double bond between C-3 and C-15. Like **1–3**, the similar ROESY correlations observed (Fig. 3) and the biosynthetic consideration supported the absolute configuration of the BPAP unit of **4** to be 1*S*, 5*R*, 7*S*, and 8*R*. Considering the fact that the configuration of **2** has been assigned to be 1*S*, 3*Z*, 5*R*, 7*S*, 8*R*, and 22*R*, it was concluded that **4** had the 1*S*, 3*Z*, 5*R*, 7*S*, 8*R*, and 22*S* configurations. Hence, **3** and **4** were defined to be a pair of *cis–trans* isomers in an approximate 5:1 ratio as determined by ¹H NMR experiments.

Hyperelamine A (**5**) exhibited a molecular formula of $C_{34}H_{45}NO_3$ with 13 degrees of unsaturation, according to the HRESIMS data ($m/z$ 538.3294 [M + Na]⁺, calcd. for 538.3292). Comparison of its 1D NMR data (Table 2) with those of **1**

suggested the absence of an α-aminopropionitrile side chain attached to C-15 of the PPAP architecture, which was replaced by an amino group in **5**. Detailed analysis of the 2D NMR spectra of **5** allowed the identification of its planar structure as shown (Fig. 1). Furthermore, the ROESY spectrum of **5** showed similar cross-peaks (Fig. 3) to those of **1**, except for the α-aminopropionitrile part. Therefore, the relative configuration of **5** was determined to be the same as that of **1**. Subsequently, an absolute configuration of 1*S*, 5*R*, 7*S*, and 8*R* for **5** was established based on ECD calculations (Fig. 5). Moreover, the single-crystal X-ray diffraction data of **5** [Flack parameter = 0.05(4), CCDC 2223398] confirmed the aforementioned configuration (Fig. 4). Thus, the absolute configuration of **5** was determined unambiguously as 1*S*, 5*R*, 7*S*, and 8*R*.

**Structural revision of garciyunnanimines A–C.** Interestingly, only three naturally occurring nitrogenous PPAPs, garciyunnanimines A–C with an imine group, have been reported in the literature so far[15]. When comparing the NMR data of **5** and garciyunnanimines A–C, it is surprising that they showed nearly identical NMR data at C-3 ($\delta_C$ 110.4–111.1) and C-15 ($\delta_C$ 171.5–172.0). However, a noticeable structural difference was observed for the C-3–C-15–N system between them (an enamine group in **5** vs. an imine group in garciyunnanimines A–C). It becomes apparent that one of the above structural assignments is incorrect. Although crystallographic data of **5** were obtained in the present study, it remains challenging to distinguish whether a C-15–N or C-15=N bond exists. To further define the structural correctness of **5**, a ¹H–¹⁵N HSQC experiment for **5** was performed. The spectrum displayed obvious cross-peaks from two protons at $\delta_H$ 12.08 and 6.09 to the nitrogen signal at $\delta_N$ 126.4 ascribed to a typical primary amine group directly attached to a double bond, which demonstrated the presence of an enamine moiety in **5**. Hence, these evidences suggested that the structures of garciyunnanimines A–C should be revised to their corresponding PPAP enamine derivatives (Supplementary Fig. S13).

**Synthesis of 1–5 from hyperelatone A (1a).** In order to corroborate the absolute configurations of **1–5** and explore their nitrogen source. A concise synthesis of **1–5** was conducted using the proposed precursor **1a**, which is also the most abundant PPAP in the extract of *H. elatoides*. First, an (*S*)-α-aminopropionitrile was synthesized from *N*-Boc-L-alanine (Supplementary Fig. S14). Subsequently, **1a** was reacted with (*S*)-α-aminopropionitrile in HOAc at 80 °C, resulting in the formation of two pairs of *cis-trans* isomers [22*R* configuration for **1** and **2** (16% yield) and 22*S* configuration for **3** and **4** (27% yield)] and **5** (23% yield) (Fig. 6a). After purification, the synthetic structures exhibited complete correspondence with the natural PPAPs **1–5**, as evidenced by a comparison of their NMR data. The (22*S*)-α-aminopropionitrile group in **3** and **4** could be racemized through a series of isomerizations under HOAc

| No. | $\delta_C$ | $\delta_H$ |
|---|---|---|
| | **Table 2** $^1$H (800 MHz) and $^{13}$C NMR (200 MHz) spectroscopic data for compound 5 in CDCl$_3$ ($\delta$ in ppm, $J$ in Hz). | |
| 1 | 69.1, C | |
| 2 | 194.3, C | |
| 3 | 111.1, C | |
| 4 | 198.7, C | |
| 5 | 59.3, C | |
| 6a | 41.4, CH$_2$ | 2.25, br d (14.2) |
| 6b | | 1.91, dd (14.2, 7.2) |
| 7 | 39.8, CH | 1.70, m |
| 8 | 51.1, C | |
| 9 | 211.5, C | |
| 10a | 26.3, CH$_2$ | 2.55, dd (13.7, 8.7) |
| 10b | | 2.50, br d (13.7) |
| 11 | 120.8, CH | 5.04, m |
| 12 | 133.4, C | |
| 13 | 26.1, CH$_3$ | 1.75, s |
| 14 | 18.0, CH$_3$ | 1.54, s |
| 15 | 171.7, C | |
| 16 | 137.6, C | |
| 17 | 126.7, CH | 7.25, d (7.5) |
| 18 | 128.5, CH | 7.39, t (7.5) |
| 19 | 130.3, CH | 7.46, t (7.5) |
| 20 | 128.5, CH | 7.39, t (7.5) |
| 21 | 126.7, CH | 7.25, d (7.5) |
| 22 | 18.4, CH$_3$ | 1.34, s |
| 23a | 28.5, CH$_2$ | 2.18, m |
| 23b | | 1.98, m |
| 24 | 124.9, CH | 4.90, t (7.0) |
| 25 | 132.3, C | |
| 26 | 25.9, CH$_3$ | 1.68, s |
| 27 | 17.8, CH$_3$ | 1.51, s |
| 28a | 35.9, CH$_2$ | 1.36, m |
| 28b | | 1.19, m |
| 29a | 22.4, CH$_2$ | 1.83, m |
| 29b | | 1.80, m |
| 30 | 124.0, CH | 5.02, t (7.3) |
| 31 | 131.7, C | |
| 32 | 25.7, CH$_3$ | 1.65, s |
| 33 | 17.6, CH$_3$ | 1.55, s |
| 34 | 19.1, CH$_3$ | 1.13, s |
| -NH$_2$ | | 12.08, br s |
| | | 6.09, br s |

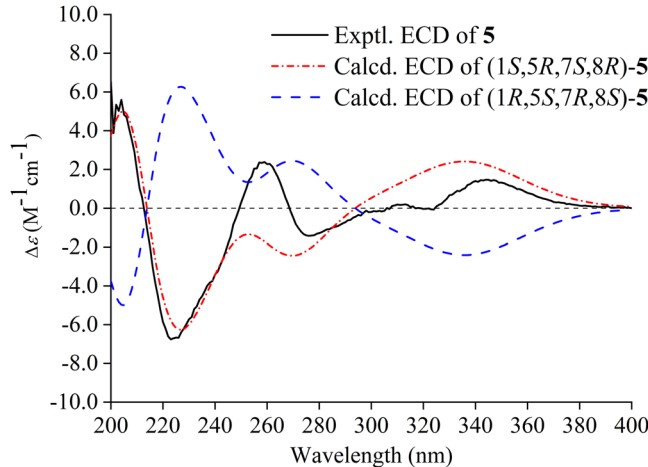

**Fig. 5 Experimental and calculated ECD spectra of compound 5.** Red and blue lines indicate the calculated ECD spectra (200–400 nm) obtained at the PBE0/def2–TZVP level in MeOH; the black line indicates the experimental ECD spectrum in MeOH.

from *Hypericum* species, compounds **1–5** were fortunately isolated and then evaluated for their anti-neuroinflammatory activity, with quercetin (IC$_{50}$ = 6.50 ± 1.13 μM) serving as a positive control. The results revealed that compounds **3/4** and **5** displayed inhibitory effects on nitric oxide (NO) production in lipopolysaccharides (LPS)-stimulated BV-2 microglial cells, with IC$_{50}$ values of 23.9 ± 1.89 and 14.4 ± 1.42 μM, respectively. In addition, none of these compounds exhibited obvious cytotoxicity to BV-2 cells at the concentrations tested. Interestingly, compound **5** showed an inhibitory effect, whereas **1a** had no effect[13], indicating that the presence of a primary enamine group attached to the C-3 position of the bicyclo[3.3.1]nonane-2,4,9-trione core could be more advantageous for this activity compared to a normal acyl group located at the C-3 position. Among these nitrogenous PPAPs, **3/4** exhibited stronger activity than **1/2** (IC$_{50}$ > 30 μM). It suggests that the absolute configuration of the α-aminopropionitrile moiety could exert an influence on this activity.

**Suppression of TLR-4/NF-κB signaling pathways by compound 5 in BV-2 cells.** Western blotting experiments were performed to clarify the underlying molecular mechanism of compound **5**. As shown in Fig. 7, compound **5** significantly suppressed the LPS-induced protein expression of inducible nitric oxide synthase (iNOS) and cyclooxygenase-2 (COX-2), which are two major markers of inflammation as well as two key mediators involved in the synthesis of NO and prostaglandins. Nuclear factor kappa-B (NF-κB) is a crucial transcriptional factor in the regulation of iNOS and COX-2 expression. Compared with the LPS group, **5** also remarkably down-regulated the expression of toll-like receptor-4 (TLR-4) and phospho-IκB kinase α (p-IKKα) and reduced nuclear/cytoplasmic NF-κB p65 ratio in a dose-dependent manner, indicating its inhibition of TLR-4/NF-κB signaling. Furthermore, the results of immunofluorescence assay showed that the preincubation with **5** blocked the LPS-induced nuclear translocation of NF-κB, which verified the inhibitory effect of compound **5** on NF-κB signaling activation.

### Conclusion

Five nitrogenous PPAPs, hyperelanitriles A–D (**1–4**) and hyperelamine A (**5**), were isolated from *H. elatoides*. Compounds **1–4** represent a new class of acylphloroglucinols possessing an α-aminonitrile moiety, while compound **5** is an unusual enamine-containing PPAP. We have successfully accomplished the

condition (Fig. 6b). The possible racemization mechanism of C-22 involved the formation of a key intermediate **ii** via N≡C protonation (**i**), leading to the generation of **1** and **2**. Compound **5** was presumably formed from **1** and **3** by the elimination of the propionitrile moiety via an enolic intermediate **iii** (Fig. 6c). Based on the aforementioned synthesis, it is reasonable to consider that the α-aminopropionitrile side chain of compounds **1–4** may be derived biogenetically from L-alanine. Furthermore, a plausible biosynthetic pathway for **1–5** is presented in Supplementary Fig. S15. Our findings have provided valuable inspiration to further modify and optimize the PPAPs by binding to the C-15 site.

### Anti-neuroinflammatory activity.

Neuroinflammation has been demonstrated to be one of the key contributors to the pathogenesis of progressive neurodegenerative diseases, most of which have no currently available drugs to treat. Therefore, the discovery and development of potential agents targeting neuroinflammation could be an important and promising therapeutic strategy for neurodegenerative diseases. In the course of our ongoing investigations of structurally novel and anti-neuroinflammatory natural products

**Fig. 6 Synthesis of compounds 1–5. a** Synthesis of **1–5** from hyperelatone A (**1a**). **b** Possible racemization mechanism of α-aminopropionitrile group. **c** Plausible formation mechanism of **5**.

synthesis of **1–5** and proposed their biosynthetic pathway. Moreover, compounds **3/4** and **5** exhibited inhibitory activity against LPS-activated NO production in BV-2 cells. Further mechanistic studies uncovered that compound **5** suppressed the neuroinflammation probably through the regulation of the TLR-4/NF-κB signaling pathway.

## Methods

**General experimental procedures**. Melting points were measured on an MP420 auto melting point apparatus (Hanon Advanced Technology Group Co., Ltd., Jinan, China) and were uncorrected. Optical rotations were determined on an Auton Paar MCP300 automatic polarimeter (Anton Paar GmbH, Graz, Austria). UV and ECD spectra were measured on an Applied Photophysics chirascan spectrometer (Applied Photophysics Ltd, Surrey, UK). IR spectra were recorded on a Bruker Tensor 27 FT-IR spectrometer with KBr disks (Bruker Corporation, Karlsruhe, Germany). 1D and 2D NMR spectra were obtained using a Bruker Avance 800 MHz spectrometer with tetramethylsilane as an internal standard (Bruker Corporation). HRESIMS were recorded on a Thermo Scientific Q Exactive mass spectrometer (Thermo Fisher Scientific, MA, USA). X-ray crystallographic data were collected on a Bruker Smart Apex II diffractometer equipped with graphite-monochromatic Cu Kα radiation ($\lambda = 1.54178$ Å) for compound **1** and a Bruker D8 VENTURE system with PHOTON II CPAD detector and a Ga-target liquid MetalJet D2 Plus X-ray source ($\lambda = 1.34139$ Å) for compound **5**. Semipreparative HPLC was performed on an Agilent 1100 series instrument (Agilent Technology, USA) equipped with a quaternary pump, a vacuum degasser, an autosampler, a diode array detector, and YMC Packed $C_{18}$ columns (5 μm, $250 \times 10.0$ and $150 \times 4.6$ mm, YMC Co., Ltd., Kyoto, Japan). Column chromatography (CC) was performed on silica gel (100–200 and 200–300 mesh, Yantai Jiangyou Silica Gel Development Co., Ltd., Yantai, China) and reversed-phase (RP)-$C_{18}$ resins (YMC Gel ODS-A-HG, 50 μm particle size, YMC Co., Ltd.). Thin-layer chromatography (TLC) was performed on silica gel HSGF254 (Yantai Jiangyou Silica Gel Development Co., Ltd.) and RP-18 F254S plates (Merck KGaA, Darmstadt, Germany).

**Plant material**. Aerial parts of *H. elatoides* R. Keller (Hypericaceae) were collected in Yihuqiao Town, Zhangxian County, Dingxi City, Gansu Province, China, in July 2019. The plant was identified by Professor Zai-Min Jiang, College of Life Sciences, Northwest A&F University. A voucher specimen (No. Yan20190721-1) was deposited in the herbarium of Northwest A&F University, Yangling, China.

**Extraction and isolation**. Dried aerial parts of *H. elatoides* (10.0 kg) were powdered and extracted with MeOH (7 × 50 L, 4 h each) under reflux conditions (55 °C). After removing the solvent under reduced pressure, the extract obtained (2.5 kg) was suspended in distilled water (5.0 L) and then partitioned with petroleum ether (4 × 5.0 L) to yield a petroleum ether-soluble fraction (537.0 g). The petroleum ether fraction was fractionated by silica gel CC and eluted with a gradient mixture of petroleum ether–$Me_2CO$ (80:1, 60:1, 40:1, 20:1, 10:1, 5:1, 2:1, 0:1, v/v) to afford 10 fractions (Frs. A–J). Fr. B was further subjected to silica

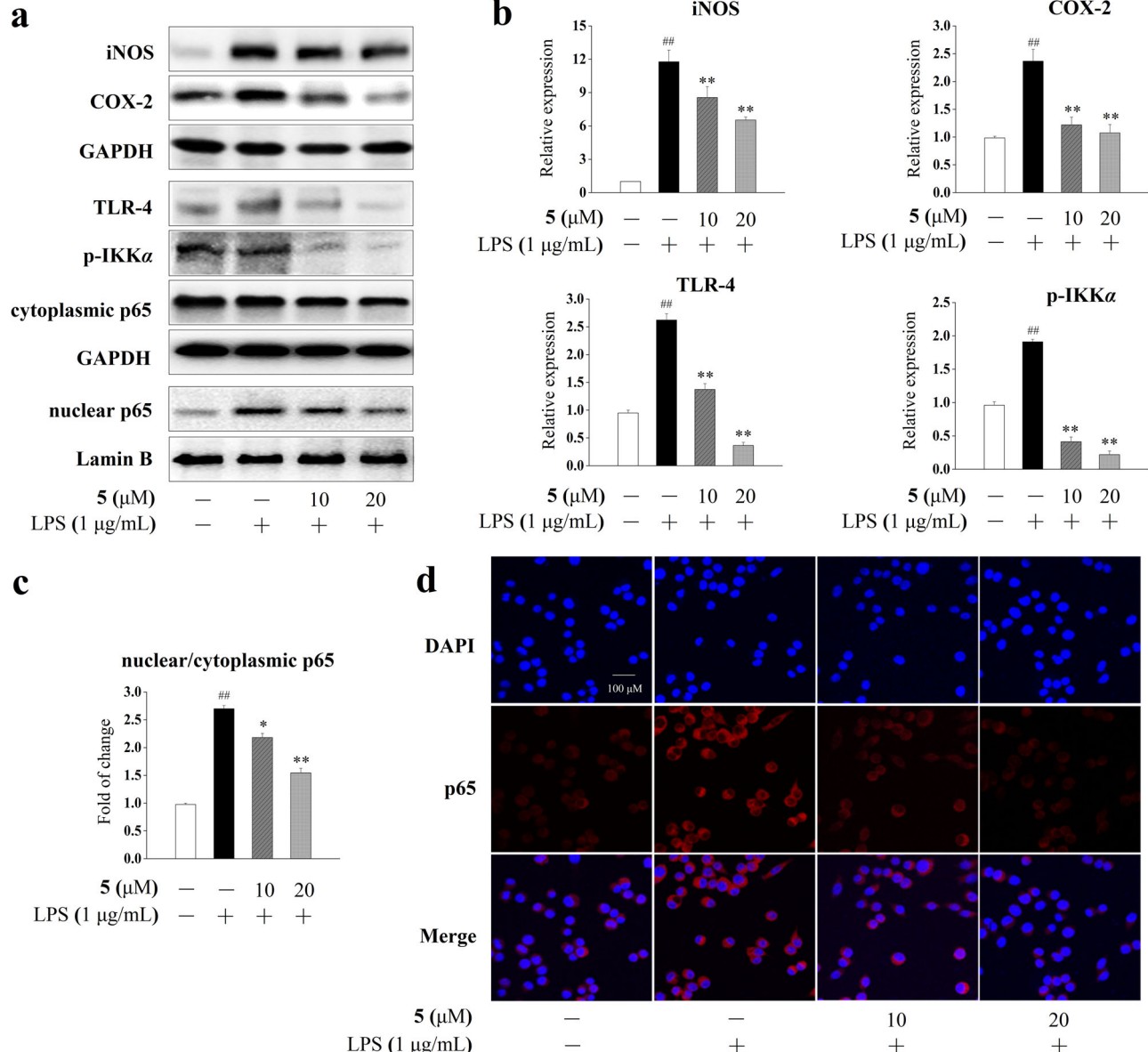

**Fig. 7 Effect of compound 5 on TLR-4/NF-κB pathway activation in BV-2 cells. a** Representative Western blots of iNOS, COX-2, TLR-4, p-IKKα, nuclear NF-κB p65, and cytoplasmic NF-κB p65. **b** Relative protein expressions of iNOS, COX-2, TLR-4, and p-IKKα. Data were presented as mean ± SD ($n = 3$ independent experiments). The significances of the intergroup differences were evaluated by one-way analysis of variance using GraphPad Prism 8.0. [##]$P < 0.01$ vs. the blank group, [**]$P < 0.01$ and [*]$P < 0.05$ vs. the LPS group. **c** Densitometric analyses of the nuclear/cytoplasmic NF-κB p65. **d** Representative immunofluorescence images of NF-κB p65 nuclear translocation (scale bar = 100 μm). The distribution of NF-κB p65 was detected by staining with an anti-p65 subunit antibody (red). Nuclei were stained in blue (DAPI).

gel CC (petroleum ether–EtOAc, 50:1, 30:1, 20:1, 5:1, v/v) to yield seven fractions (Frs. B1–B7). Fr. B5 was separated using RP-C$_{18}$ CC (Me$_2$CO–H$_2$O, 3:1, 4:1, 5:1, 8:1, 10:1, 1:0, v/v) to produce 12 subfractions (Frs. B5.1–B5.12). Fr. B5.7 was purified successively by semipreparative HPLC (MeOH–H$_2$O, from 70:30 to 100:0 in 30 min) and silica gel CC (petroleum ether–EtOAc, 10:1, v/v) to afford compounds **1/2** (5.5 mg) and **3/4** (4.0 mg). Fr. B6 was subjected to RP-C$_{18}$ CC (Me$_2$CO–H$_2$O, 3:1, 4:1, 5:1, 8:1, 1:0, v/v) to furnish nine subfractions (Frs. B6.1–B6.9). Fr. B6.5 was further purified by semipreparative HPLC (MeCN–H$_2$O, from 70:30 to 100:0 in 30 min) to provide compound **5** (3.8 mg).

*Hyperelanitrile A (1).* Colorless crystals; m.p. 144–147 °C; $[\alpha]_D^{25}$ + 91.1 (*c* 0.05, MeOH); UV (MeOH) $\lambda_{max}$ (log $\varepsilon$) 248 (4.11), 320 (4.38) nm; ECD (MeOH) $\lambda_{max}$ ($\Delta\varepsilon$) 205 (+15.43), 226

(−10.62), 261 (+3.59), 284 (−2.30), 318 (−5.67) nm; IR (KBr) $\nu_{max}$ 3407, 3125, 2925, 2861, 2392, 2308, 1722, 1658, 1596, 1543, 1449, 1386, 1313, 1263, 1097, 1038, 806, 691 cm$^{-1}$; $^1$H NMR (CDCl$_3$, 800 MHz) and $^{13}$C NMR (CDCl$_3$, 200 MHz) data, (see Table 1); (+)-HRESIMS *m/z* 591.3559 [M + Na]$^+$ (calcd. for C$_{37}$H$_{48}$N$_2$O$_3$Na, 591.3557).

*Hyperelanitrile B (2).* Colorless oil; $[\alpha]_D^{25}$ + 91.1 (*c* 0.05, MeOH); UV (MeOH) $\lambda_{max}$ (log $\varepsilon$) 248 (4.11), 320 (4.38) nm; ECD (MeOH) $\lambda_{max}$ ($\Delta\varepsilon$) 205 (+15.43), 226 (−10.62), 261 (+3.59), 284 (−2.30), 318 (−5.67) nm; IR (KBr) $\nu_{max}$ 3407, 3125, 2925, 2861, 2392, 2308, 1722, 1658, 1596, 1543, 1449, 1386, 1313, 1263, 1097, 1038, 806, 691 cm$^{-1}$; $^1$H NMR (CDCl$_3$, 800 MHz) and $^{13}$C NMR (CDCl$_3$, 200 MHz) data (see Table 1); (+)-HRESIMS *m/z* 591.3559 [M + Na]$^+$ (calcd. for C$_{37}$H$_{48}$N$_2$O$_3$Na, 591.3557).

*Hyperelanitrile C (3)*. Colorless oil; $[\alpha]_D^{25}$ +28.3 (*c* 0.10, MeOH); UV (MeOH) $\lambda_{max}$ (log $\varepsilon$) 248 (3.92), 321 (4.14) nm; ECD (MeOH) $\lambda_{max}$ ($\Delta\varepsilon$) 202 (+3.30), 227 (−10.19), 259 (+0.33), 275 (−3.09), 313 (+2.60), 346 (+2.19) nm; IR (KBr) $\nu_{max}$ 3129, 2929, 2399, 2307, 1723, 1658, 1542, 1394, 1313, 1049, 807, 673 cm$^{-1}$; $^1$H NMR (CDCl$_3$, 800 MHz) and $^{13}$C NMR (CDCl$_3$, 200 MHz) data (see Table 1); (+)-HRESIMS *m/z* 591.3558 [M + Na]$^+$ (calcd. for C$_{37}$H$_{48}$N$_2$O$_3$Na, 591.3557).

*Hyperelanitrile D (4)*. Colorless oil; $[\alpha]_D^{25}$ + 28.3 (*c* 0.10, MeOH); UV (MeOH) $\lambda_{max}$ (log $\varepsilon$) 248 (3.92), 321 (4.14) nm; ECD (MeOH) $\lambda_{max}$ ($\Delta\varepsilon$) 202 (+3.30), 227 (−10.19), 259 (+0.33), 275 (−3.09), 313 (+2.60), 346 (+2.19) nm; IR (KBr) $\nu_{max}$ 3129, 2929, 2399, 2307, 1723, 1658, 1542, 1394, 1313, 1049, 807, 673 cm$^{-1}$; $^1$H NMR (CDCl$_3$, 800 MHz) and $^{13}$C NMR (CDCl$_3$, 200 MHz) data (see Table 1); (+)-HRESIMS *m/z* 591.3558 [M + Na]$^+$ (calcd. for C$_{37}$H$_{48}$N$_2$O$_3$Na, 591.3557).

*Hyperelamine A (5)*. Colorless crystals; m.p. 145−148 °C; $[\alpha]_D^{25}$ +44.3 (*c* 0.11, MeOH); UV (MeOH) $\lambda_{max}$ (log $\varepsilon$) 257 (3.77), 316 (3.80) nm; ECD (MeOH) $\lambda_{max}$ ($\Delta\varepsilon$) 204 (+5.60), 223 (−6.76), 259 (+2.38), 276 (−1.43), 313 (+0.20), 345 (+1.47) nm; IR (KBr) $\nu_{max}$ 3142, 2397, 2308, 1643, 1581, 1399, 1264, 1082, 804, 668 cm$^{-1}$; $^1$H NMR (CDCl$_3$, 800 MHz) and $^{13}$C NMR (CDCl$_3$, 200 MHz) data (see Table 2); (+)-HRESIMS *m/z* 538.3294 [M + Na]$^+$ (calcd. for C$_{34}$H$_{45}$NO$_3$Na, 538.3292).

**NMR and ECD calculations**. The methods and details for NMR and ECD calculations are provided in Supplementary Methods 1.1 and 1.2.

**X-ray crystallographic analysis of 1 and 5**. Colorless single crystals of compounds 1 and 5 were obtained in a solution of CHCl$_3$–MeOH (1:10) and MeOH (100%) by slow evaporation at 4 °C, respectively. Their structures were solved by SHELXT (version 2018/2) and refined by full-matrix least-squares procedures using the SHELXL program. Crystallographic data for 1 and 5 were deposited in the Cambridge Crystallographic Data Center with the deposition numbers CCDC 2151888 and 2223398, respectively. These data can be obtained free of charge via the Internet at www.ccdc.cam.ac.uk.

*Crystal data of 1*. C$_{37}$H$_{48}$N$_2$O$_3$, $M$ = 568.77, orthorhombic, space group $P2_12_12_1$ (no. 19), $a$ = 8.5213(4) Å, $b$ = 11.2417(6) Å, $c$ = 34.6213(17) Å, $\alpha = \beta = \gamma = 90°$, $V$ = 3316.5(3) Å$^3$, $Z$ = 4, $T$ = 296.15 K, $\mu$(Cu K$\alpha$) = 0.557 mm$^{-1}$, $D_{calc}$ = 1.139 g/cm$^3$, 28,235 reflections measured (8.27° ≤ 2θ ≤ 141.904°), 6277 unique ($R_{int}$ = 0.0356, $R_{sigma}$ = 0.0273), which were used in all calculations. The final $R_1$ was 0.0596 [$I > 2\sigma(I)$] and w$R_2$ was 0.1655 (all data). The goodness of fit on $F^2$ was 1.051. Flack parameter = 0.14(7).

*Crystal data of 5*. C$_{34}$H$_{45}$NO$_3$, $M$ = 515.71, trigonal, space group $P3_1$ (no. 144), $a = b$ = 25.9064(6) Å, $c$ = 11.8303(3) Å, $\alpha = \beta = 90°$, $\gamma = 120°$, $V$ = 6876.1(4) Å$^3$, $Z$ = 9, $T$ = 200.15 K, $\mu$(Ga K$\alpha$) = 0.348 mm$^{-1}$, $D_{calc}$ = 1.121 g/cm$^3$, 324,792 reflections measured (6.856° ≤ 2θ ≤ 123.79°), 21,507 unique ($R_{int}$ = 0.0627, $R_{sigma}$ = 0.0323), which were used in all calculations. The final $R_1$ was 0.0413 [$I > 2\sigma(I)$] and w$R_2$ was 0.1010 (all data). The goodness of fit on $F^2$ was 1.047. Flack parameter = 0.05(4).

**Procedure for the synthesis of 1–5**. The (*S*)-α-aminopropanenitrile was synthesized from *N*-Boc-L-alanine using methods reported in the literature[16,17]. Briefly, *N*-Boc-L-alanine served as the starting material and reacted with ethyl chloroacetate and aqueous ammonia to produce a primary amide intermediate, which was pure enough for use in the subsequent step without additional purification. The nitrile intermediate was obtained by dehydration using trifluoroacetic anhydride in anhydrous pyridine as solvent. Subsequently, the desired (*S*)-α-aminopropanenitrile was obtained after deprotection of the Boc group. Finally, the proposed precursor 1a was reacted with (*S*)-α-aminopropionitrile under HOAc condition according to the following detailed procedures. A 25 mL round-bottom flask equipped with a magnetic stir bar was charged with hyperelatone A (1a, 100 mg), (*S*)-α-aminopropanenitrile (0.67 g, 0.95 mmol, 5 equiv.), and acetic acid (10.88 mL, 190 mmol, 1000 equiv.). The reaction mixture was refluxed at 80 °C under magnetic stirring for 10 h. After cooling to room temperature, the reaction solution was evaporated to dryness. The resulting residue was subjected to analysis via TLC and HPLC, which demonstrated successful conversion into compounds 1–5. The products were further purified by semipreparative HPLC to afford compounds 1/2, 3/4, and 5, respectively.

**Anti-neuroinflammatory assay**. BV-2 microglial cells were purchased from Peking Union Medical College Cell Bank (Beijing, China) and cultured in Dulbecco's modified Eagle's medium (DMEM) supplemented with 10% fetal bovine serum (FBS) and 1% penicillin/streptomycin (Solarbio Science & Technology Co., Ltd., Beijing, China) at 37 °C in a 5% CO$_2$ incubator. The cell viability of the cultured cells was determined using a 3-(4,5-dimethylthiazol-2-yl)-2,5-diphenyl tetrazolium bromide (MTT) assay. The NO concentration was detected by the Griess reagent (Beyotime Biotechnology, Shanghai, China). Briefly, BV-2 cells were seeded at the density of $1.5 \times 10^5$ cells per mL in 96-well plates and treated individually with five different concentrations (1.25, 2.5, 5.0, 10.0, and 20.0 μM) of each test compound and LPS (1.0 μg mL$^{-1}$) for 24 h. Next, 50 μL of cell-free supernatant was allowed to react with an equal volume of Griess reagent at room temperature in the dark. After 15 min, the absorbance was measured at 540 nm using a microplate reader.

**Western blotting**. BV-2 cells were treated with DMSO or compound 5 (10 and 20 μM) and then stimulated with or without 1 μg mL$^{-1}$ LPS for 24 h. Proteins were extracted using radio-immunoprecipitation assay (RIPA) lysis buffer (Beyotime Biotechnology) containing protease and phosphatase inhibitor cocktails. The concentrations of proteins were determined by a bicinchoninic acid (BCA) protein assay kit (Shanghai Epizyme Biomedical Technology Co., Ltd., China). The proteins were isolated and subjected to electrophoresis on 7.5% sodium dodecyl sulfate–polyacrylamide gels (SDS–PAGE) and transferred onto a nitrocellulose filter membrane. After 1 h incubation in 5% bovine serum albumin (BSA) in the mixture of Tris-buffered saline and Tween-20 (TBST), the membranes were probed with desired primary antibodies at 4 °C for 12 h and subsequently detected by specific secondary antibodies for 1 h. The following primary antibodies were used: iNOS (Rabbit monoclonal antibody, 1:1000, ABclonal Technology Co., Ltd., China), COX-2 (Rabbit polyclonal antibody, 1:1000, Proteintech Group, Inc., USA), TLR-4 (Rabbit polyclonal antibody, 1:1000, ABclonal Technology Co., Ltd.), p-IKKα (Rabbit polyclonal antibody, 1:1000, Signalway Antibody, USA), NF-κB (Mouse monoclonal antibody, 1:1000, Novus Biologicals, CO, USA), Lamin B (Rabbit polyclonal antibody, 1:1000, Proteintech Group, Inc.), and GAPDH (Rabbit polyclonal antibody, 1:5000, Proteintech Group, Inc.). Horseradish peroxidase (HRP)-conjugated secondary antibodies used were goat anti-rabbit IgG (1:2000, Cell Signaling Technology, USA) and goat anti-mouse IgG (1:2000, Proteintech Group, Inc.). The immunoreactive signals were detected using the Alliance Q9 Advanced system (Uvitec Ltd., Cambridge, UK).

**Immunofluorescence**. Immunofluorescence staining was performed as described previously[18,19]. After treatments with compound **5** (10 and 20 μM) and LPS (1 μg mL$^{-1}$) for 24 h, the cell-seeded glass coverslips were fixed with 4% cold paraformaldehyde for 20 min and permeabilized with 0.3% Triton X-100 for 30 min. Then, the cells were blocked with 5% BSA (in PBS) for 1 h at room temperature and incubated with a primary antibody specific to the NF-κB p65 subunit (Novus, USA) for 3 h at 37 °C, followed by a secondary antibody labeled with FITC (1:500) for 1 h at room temperature in the dark. Afterward, the cells were counterstained with DAPI (5 μg mL$^{-1}$ in PBS) for 10 min at room temperature. Finally, the coverslips were washed and sealed. Images were observed using a laser scanning confocal microscopy (Leica, Solms, Germany) with the excitation/emission wavelengths of 492/520 nm for FITC and 360/450 nm for DAPI.

**Reporting summary**. Further information on research design is available in the Nature Portfolio Reporting Summary linked to this article.

## Data availability
The authors declare that all the data supporting the findings of this study are available within the Article itself and its Supplementary Information. For computational details, see Supplementary Methods 1.1 and 1.2, Supplementary Tables S1–S7, and Supplementary Figs. S1–S12. The 1D and 2D NMR, HRESIMS, UV, ECD, and IR spectra of compounds **1**–**5** are available as Supplementary Data 1. The X-ray crystallographic data for compounds **1** and **5** have been deposited at the Cambridge Crystallographic Data Center (CCDC), under deposition numbers CCDC 2151888 and 2223398, respectively. These data can be obtained free of charge from The Cambridge Crystallographic Data Centre via www.ccdc.cam.ac.uk/data_request/cif. The CIF files of CCDC 2151888 and 2223398 are also included as Supplementary Data 2 and 3.

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

## Acknowledgements
This work was supported by the National Natural Science Foundation of China (31800291).

## Author contributions
X.-T.Y. and J.-M.G. conceived, designed, and supervised the project. J.-Y.X. conducted most of the phytochemical and biological experiments. P.L. performed the synthetic work of **1**–**5**. X.-T.Y. analyzed the spectroscopic data and discussed the results. All authors contributed to the writing of the manuscript.

## Competing interests
The authors declare no competing interests.
