## [Peer Review File · Communications Chemistry]

Reviewers' comments:

Reviewer #1 (Remarks to the Author):

Reviewer Comments: Discovery and synthesis of hyperelanitriles A–D, α -aminopropionitrile-containing 1 polycyclic polyprenylated acylphloroglucinols from *Hypericum elatoides*

The major claims of the paper are the isolation and full spectroscopic characterization of 5 novel compounds in the PPAP family, four of which contain a previously unobserved aminopropionitrile moiety. The synthesis of these compounds has also been demonstrated, generating a mixture of all five isomers. The authors have a substantial body of work in this area of study and their characterization was relatively easy to follow. The paper will be of interest to those in the field and to those looking for natural product scaffolds with potential antineuroinflammatory activity. The identification of this class of compounds as potential antineuroinflammatory agents is not new, but again, these specific compounds are new. The authors do not state whether the activity observed is better than similar compounds (which it is not based on their own recent work, <https://doi.org/10.1021/acs.jnatprod.3c00226>). I believe this should be mentioned in the paper, especially since it is the work of the authors themselves and was recently published.

As I am not an expert in running these particular biological assays, so I cannot comment on other experiments that would benefit the paper in this particular area. The experiments conducted are well explained and provide appropriate context which I believe would be reproducible by other researchers. I also cannot comment on the specific computational analysis, although it is consistent with other methods reported in the literature in this area.

The most critical comment I have is that although it is mentioned that compounds 1 and 2, as well as 3 and 4, are mixtures, the supporting 1D NMR data, experimental ECD suggests only one when it can clearly be seen from the NMR data that other peaks from the other isomer are observable in the spectrum. I believe the author was just trying to highlight which peaks are from which isomer, but this is not necessarily clear and the fact that they are mixtures should be addressed in the figure titles. It may be nice to have larger, full page ¹H NMR figures that clearly indicate these are mixtures and colour code the peaks associated with each compound. There is a melting point provided for 1, so I would assume clean data could be obtained free from isomer 2. Is this data not available? An LC-MS trace of these compounds would also be an ideal accompaniment to the NMR data of the mixture to show that there are 2 isomers with the same mass.

The authors comment on the potential incorrect structural analysis of a set of three previously published molecules in reference 10. (Zheng, D.; Zhang, H.; Zheng, C. W.; Lao, Y. Z.; Xu, D. Q.; Xiao, L. B.; Xu, H. X. Garcinyunnanamines A–C, novel cytotoxic polycyclic polyprenylated acylphloroglucinol imines from *Garcinia yunnanensis*. *Org. Chem. Front.* 2017, 4, 2102–2108.) I would like to suggest that the authors contact the authors of the aforementioned paper if they haven't done so to discuss this difference in structural identification.

A few minor comments or correction requests for the authors:

The methylene protons on C6, C10, C32 and C26 for compound 1 are not mentioned in the paragraph at line 73.

Line 131 should be "in 2" not "in 3".

Line 94 and line 134 refer to ref 15 after the Δ , which I believe should be ref 5. There is no reference 15 listed in the reference section.

Line 353, add "1a" after hyperelactone A for clarity.

Yields for synthesis of 1-5 is not reported and should be.

Type of IR spectrum not reported on spectrum.

Overall, the work has been completed for publication, although I would suggest the authors clean up their spectral data reporting before publication, especially if isolation of each compound individually is not possible.

My suggestion to the editors would be acceptance with minor revisions.

Reviewer #2 (Remarks to the Author):

1. This manuscript mainly reported five rare nitrogenous polycyclic polyprenylated acylphloroglucinols and their structures were determined by spectroscopic data, ECD and NMR calculations, and X-ray crystallography. A semi-synthesis of compounds 1-5 was accomplished. Based on their semi-synthesis, these compounds are unstable under acid condition. However, during the isolation procedure, no acid solvent was involved, suggesting whether inseparable compounds 1, 2 and 3, 4 are instability due to the presence of water or solvent containing free hydrogen. I suggest that author uses chiral column to analyze compounds 1 and 2, as well as 3 and 4. Then their stability should be determined using HPLC.

2. In addition, garciyunnanimines A-C with an imine group were isolated and their structures contain C=NH and enol, whereas, author found the similar compounds with enamine and ketone. I think two types could be present. Thus, author also should consider the stable of compound 5. If it is unstable, their ratio should be determined and analyze its thermodynamic stability. Meanwhile, the stable configuration of 5 can be determined by the quantum chemical calculation method.

Reviewer #3 (Remarks to the Author):

This manuscript described five nitrogenous polycyclic polyprenylated acylphloroglucinols (PPARs) and their anti-neuroinflammatory activity. Their structures were determined by spectroscopic analysis, ECD and NMR calculations, and X-ray. The structures with α -aminonitrile moiety were interesting. This

manuscript could be accepted after the following issues are solved.

1. For the structural elucidation for compound 1, the evidence for the presence of CN group was not enough. The IR spectrum should have a characterized peak at 2210 cm^{-1} . I carefully checked the IR data of the five compounds in Experiments part, no 2210 cm^{-1} was provided. Please recheck the IR spectrum and rewrite the structural elucidation for compound 1.
2. In Fig.3, some ROESY information are not needed, for example the two hydrogens of NH_2 in compound 5. Please recheck all the ROESY correlations.

List of Responses

Journal: **Communications Chemistry**

Manuscript ID: **COMMSCHEM-23-0381**

I. Comments from Reviewer 1

The major claims of the paper are the isolation and full spectroscopic characterization of 5 novel compounds in the PPAP family, four of which contain a previously unobserved aminopropionitrile moiety. The synthesis of these compounds has also been demonstrated, generating a mixture of all five isomers. The authors have a substantial body of work in this area of study and their characterization was relatively easy to follow. The paper will be of interest to those in the field and to those looking for natural product scaffolds with potential antineuroinflammatory activity. The identification of this class of compounds as potential antineuroinflammatory agents is not new, but again, these specific compounds are new. The authors do not state whether the activity observed is better than similar compounds (which it is not based on their own recent work, <https://doi.org/10.1021/acs.jnatprod.3c00226>). I believe this should be mentioned in the paper, especially since it is the work of the authors themselves and was recently published.

Answer: Firstly, thanks a lot for your review and positive comments. We are sorry for our oversight in omitting the discussion of the biological results concerning the structure-activity relationship with similar compounds. Based on your valuable suggestion, a recently published work (<https://doi.org/10.1021/acs.jnatprod.3c00226>) have been referenced and incorporated into the revised manuscript. The relevant discussion has been added in our manuscript (**Lines 234–240, Page 15**). Please review them below.

Lines 234–240, Page 15:

Interestingly, compound **5** showed inhibitory effect, whereas **1a** had no effect¹³, indicating that the presence of a primary enamine group attached to the C-3 position of the bicyclo[3.3.1]nonane-2,4,9-trione core could be more advantageous for this activity

compared to a normal acyl group located at the C-3 position. Among these nitrogenous PPAPs, **3/4** exhibited stronger activity than **1/2** ($IC_{50} > 30 \mu M$). It suggests that the absolute configuration of the α -aminopropionitrile moiety could exert an influence on this activity.

As I am not an expert in running these particular biological assays, so I cannot comment on other experiments that would benefit the paper in this particular area. The experiments conducted are well explained and provide appropriate context which I believe would be reproducible by other researchers. I also cannot comment on the specific computational analysis, although it is consistent with other methods reported in the literature in this area.

The most critical comment I have is that although it is mentioned that compounds **1** and **2**, as well as **3** and **4**, are mixtures, the supporting 1D NMR data, experimental ECD suggests only one when it can clearly be seen from the NMR data that other peaks from the other isomer are observable in the spectrum. I believe the author was just trying to highlight which peaks are from which isomer, but this is not necessarily clear and the fact that they are mixtures should be addressed in the figure titles. It may be nice to have larger, full page 1H NMR figures that clearly indicate these are mixtures and colour code the peaks associated with each compound.

Answer: Thanks for your helpful suggestion. The full 1D NMR spectra of a mixture of **1** and **2** and a mixture of **3** and **4**, along with their enlarged spectral figures, have been presented in the revised supplementary information file. Their figure titles have been modified to accurately reflect the spectrum of a mixture of both **1** and **2** or a mixture of both **3** and **4**. Additionally, distinct NMR peak assignments for each isomer were also shown on their spectra using different color codes. The revised figures present these new compounds with enhanced clarity compared to the previous supplementary figures. Please review the revised **Figures S16–S20** (1H NMR spectrum for a mixture of **1** and **2**), **Figures S21–S25** (^{13}C NMR spectrum for a mixture of **1** and **2**), **Figures S38–S42** (1H NMR spectrum for a mixture of **3** and **4**), and **Figures S43–S47** (^{13}C NMR

spectrum for a mixture of **3** and **4**) in the revised supplementary information. For example, the **Figures S17** and **S24** in the revised supplementary information has been updated as depicted below.

New Figure S17

New Figure S24

There is a melting point provided for **1**, so I would assume clean data could be obtained free from isomer **2**. Is this data not available? An LC-MS trace of these compounds would also be an ideal accompaniment to the NMR data of the mixture to show that there are 2 isomers with the same mass.

Answer: We appreciate your valuable suggestion. During the isolation of compounds **1–4**, pure crystals of **1** with the absolute configuration of *1S, 3E, 5R, 7S, 8R, 22R* were obtained in our study, as confirmed by single-crystal X-ray diffraction analysis. However, its NMR spectra also exhibited a mixture of both **1** and **2** in an approximate 2:1 ratio when the pure crystals of **1** were dissolved in solution. We have tried various solvents and methods to obtain the crystals of **2** from the solution, but our efforts have been unsuccessful. According to your suggestion, an LC-MS trace experiment was conducted on these compounds, revealing the presence of two pairs of *cis-trans* isomers with identical mass. This was illustrated in the following figures.

Extracted ion chromatogram (XIC) of compounds **1–5**

HRMS of compound **5** (the first peak in XIC, found 516.3472)

HRMS of a mixture of compounds **3** and **4** (the second peak in XIC, found 569.3730)

HRMS of a mixture of compounds **1** and **2** (the third peak in XIC, found 569.3734)

The authors comment on the potential incorrect structural analysis of a set of three previously published molecules in reference 10 (Zheng, D.; Zhang, H.; Zheng, C. W.; Lao, Y. Z.; Xu, D. Q.; Xiao, L. B.; Xu, H. X. Garciyunnanimines A–C, novel cytotoxic polycyclic polyprenylated acylphloroglucinol imines from *Garcinia yunnanensis*. *Org. Chem. Front.* 2017, 4, 2102–2108.). I would like to suggest that the authors contact the authors of the aforementioned paper if they haven't done so to discuss this difference in structural identification.

Answer: Thank you very much for your kind suggestion. In our study, the ^1H – ^{15}N HSQC, ^1H – ^1H COSY, and ROESY spectra of compound **5** have provided directly evidence for the presence of a typical primary enamine group. The two protons at δ_{H} 12.08 and 6.09 exhibited significant coupling with each other and also showed correlations with the nitrogen atom. The chemical shift of the nitrogen atom observed at δ_{N} 126.4 further demonstrated the presence of a C–N single bond in enamine (range from 90 to 140 ppm) rather than a C=N double bond in imines (range from 305 to 375 ppm). So, we make sure that compound **5** is correctly identified as an enamine-containing PPAP. However, the NMR spectra of garciyunnanimines A–C showed nearly identical chemical shifts at C-3 (δ_{C} 110.4–111.1) and C-15 (δ_{C} 171.5–172.0) in comparison to compound **5**, suggesting that they should possess an identical primary enamine moiety rather than an imine group. Based on the above analysis and evidences, we believe that it is necessary to revise the structures of garciyunnanimines A–C to their corresponding PPAP enamines. Additionally, in order to comply with the journal style and formatting of Communications Chemistry [a maximum of 10 display items (figures and tables) for ~5000 words], we have relocated the previous Figure 6 (structural reassignment of garciyunnanimines A–C) in the article to Supplementary Information file as supplementary **Figure S13** at **Page 22**. Please kindly review it.

Expansion of ^1H - ^{15}N HSQC spectrum of compound 5

Expansion of ^1H - ^1H COSY spectrum of compound 5

Expansion of ROESY spectrum of compound 5

A few minor comments or correction requests for the authors:

(1) The methylene protons on C-6, C-10, C-32, and C-26 for compound **1** are not mentioned in the paragraph at line 73.

Answer: Thanks for your suggestion. The ¹H NMR analysis section of compound **1** has been updated to include these methylene protons, and the structural elucidation of compound **1** has been revised to enhance its accuracy and verifiability. The revised portions were highlighted in yellow color at **lines 71–80, Pages 4–5**. Please kindly review this section once again in the revised manuscript.

Lines 72–80, Pages 4–5:

The ¹H NMR spectrum of **1**, combined with the analysis of the HSQC spectrum, revealed the presence of a NH group with a dramatic downfield proton [δ_{H} 13.33 (d, J = 8.7 Hz)] due to strong intramolecular hydrogen bonding, one unusual monosubstituted benzene moiety with five nonequivalent protons [δ_{H} 6.99 (d, J = 7.6 Hz), 7.44 (t, J = 7.6 Hz), 7.50 (t, J = 7.6 Hz), 7.54 (t, J = 7.6 Hz), and 7.19 (d, J = 7.6 Hz)], three trisubstituted olefins (δ_{H} 5.01, 4.90, and 4.81), two sp³ methines (δ_{H} 4.16 and 1.71), five sp³ methylenes (δ_{H} 2.46, 2.40, 2.22, 2.18, 1.98, 1.93, 1.82, 1.77, 1.32, 1.15), and nine methyls [δ_{H} 1.70 (s), 1.66 (s), 1.65 (s), 1.63 (d, J = 7.1 Hz), 1.55 (s), 1.51 (s), 1.40 (s), 1.35 (s), and 1.10 (s)] (Table 1).

(2) Line 131 should be “in **2**” not “in **3**”.

Answer: This error has been corrected in the revised manuscript, Thank you.

(3) Line 94 and line 134 refer to ref 15 after the Δ , which I believe should be ref 5. There is no reference 15 listed in the reference section.

Answer: We apologize for the use of the ambiguous symbol “ $\Delta^{3,15}$ ” to represent the double bond between C-3 and C-15 in our manuscript. According to your kind suggestion, all the symbols “ $\Delta^{3,15}$ ” have been replaced with appropriate word description. Please review them at **Line 93, Page 5; Line 137, Page 8; Lines 150 and**

166, Page 10; Line 208, Page 13.

Line 93, Page 5: replace “the $\Delta^{3,15}$ double bond” with “the C-3–C-15 double bond”.

Line 137, Page 8: replace “the *Z*-configured $\Delta^{3,15}$ ” with “the 3*Z*-configured double bond”.

Line 150, Page 10: replace “the 3*E* geometry of $\Delta^{3,15}$ ” with “the 3*E* geometry of double bond”.

Line 166, Page 10: replace “*cis-trans* $\Delta^{3,15}$ double bond isomers” with “*cis-trans* isomers”.

Line 208, Page 13: replace “*cis-trans* $\Delta^{3,15}$ isomers” with “*cis-trans* isomers”.

(4) Line 353, add “**1a**” after hyperelactone A for clarity.

Answer: The compound number “**1a**” has been added after hyperelactone A at **Line 388, Page 23**, in accordance with your suggestion.

(5) Yields for synthesis of **1–5** is not reported and should be.

Answer: We sincerely apologize for our oversight in reporting the yields of compounds **1–5** in the synthetic experiment. We have added these yields in the “synthesis of **1–5** from hyperelactone A (**1a**)” section at **Lines 206–209, Page 13**. Please review the added information.

Lines 206–209, Page 13:

Subsequently, **1a** was reacted with (*S*)- α -aminopropionitrile in HOAc at 80 °C, resulting in the formation of two pairs of *cis-trans* isomers [22*R* configuration for **1** and **2** (16% yield) and 22*S* configuration for **3** and **4** (27% yield)] and **5** (23% yield) (Fig. 6a).

(6) Type of IR spectrum not reported on spectrum.

Answer: Following your valuable suggestion, the detection type (film on KBr pellet)

for IR spectra have been added into the legends of IR spectra in the Supplementary Information. For example, the legend for Figure S36 has been revised to “IR spectrum (film on KBr pellet) of **1**/**2**.” Please review these corrections (**Figure S36, Page 44; Figure S58, Page 66; Figure S72, Page 80**) in the revised Supplementary Information.

Overall, the work has been completed for publication, although I would suggest the authors clean up their spectral data reporting before publication, especially if isolation of each compound individually is not possible. My suggestion to the editors would be acceptance with minor revisions.

Answer: Thank you very much for your valuable comments and suggestions again. The NMR data for compounds **1–5** have been meticulously double-checked and we can assure the precision of these data. All necessary corrections have been highlighted in yellow within both the MARKED version Manuscript and Supplementary Information. We hope that these corrections will receive your approval.

II. Comments from Reviewer 2

1. This manuscript mainly reported five rare nitrogenous polycyclic polyprenylated acylphloroglucinols and their structures were determined by spectroscopic data, ECD and NMR calculations, and X-ray crystallography. A semi-synthesis of compounds **1–5** was accomplished. Based on their semi-synthesis, these compounds are unstable under acid condition. However, during the isolation procedure, no acid solvent was involved, suggesting whether inseparable compounds **1, 2** and **3, 4** are instability due to the presence of water or solvent containing free hydrogen. I suggest that author uses chiral column to analyze compounds **1** and **2**, as well as **3** and **4**. Then their stability should be determined using HPLC.

Answer: Firstly, thank you very much for your nice review and good comments for our manuscript. During the isolation of compounds **1–4**, we have found that **1** and **2** exist as a pair of inseparable *cis-trans* isomers in an approximate 2:1 ratio in solution, both

having the same absolute configuration of *22R*. Similarly, **3** and **4** exist as a pair of inseparable *cis-trans* isomers in an approximate 5:1 ratio in solution, with the same absolute configuration of *22S*. For example, pure crystals of **1** with the *3E*-configured double bond were obtained in our experiment, as confirmed by single-crystal X-ray diffraction analysis. However, its NMR spectra also exhibited a mixture of both **1** and **2** when dissolved in NMR solvents. Nevertheless, the structures of **1–4** have been characterized unambiguously in this work due to their distinctive NMR signals observed using an 800 MHz NMR spectrometer.

Compounds **1/2** and **3/4** could be separated well using RP-C₁₈ HPLC column, whereas the mixture **1** and **2** (or **3** and **4**) only showed a single peak in the HPLC chromatogram without any possibility of individual isolation. According to your suggestion, we have analyzed the mixture of **1** and **2** as well as the mixture of **3** and **4** using chiral HPLC columns. However, each mixture still exhibited a single peak in the HPLC chromatogram. Please check the following HPLC chromatograms. Therefore, the mixture **1** and **2** (or **3** and **4**) was described as “an inseparable mixture when in solution” in this manuscript.

1.1 Chiral column: CHIRALPAK AD-H (5 μ m, 250 \times 4.6 mm, Lot No.: ADHOCE-DR344)

1.1.1 Chiral HPLC chromatogram of compounds **1** and **2** (UV 254 nm)

1.1.2 Chiral HPLC chromatogram of compounds **3** and **4** (UV 254 nm)

1.2 Chiral column: CHIRALCEL OD-H (5 μ m, 250 \times 4.6 mm, Lot No.: ODHCE-NK020)

1.2.1 Chiral HPLC chromatogram of compounds **1** and **2** (UV 254 nm)

1.2.2 Chiral HPLC chromatogram of compounds **3** and **4** (UV 254 nm)

1.3 Chiral column: CHIRALPAK IB-3 (3 μm , 250 \times 4.6 mm, Lot No.: IB3OCE-QDO11)

1.3.1 Chiral HPLC chromatogram of compounds 1 and 2 (UV 254 nm)

1.3.2 Chiral HPLC chromatogram of compounds 3 and 4 (UV 254 nm)

2. In addition, garciyunnanimines A–C with an imine group were isolated and their structures contain C=NH and enol, whereas, author found the similar compounds with enamine and ketone. I think two types could be present. Thus, author also should consider the stable of compound **5**. If it is unstable, their ratio should be determined and analyze its thermodynamic stability. Meanwhile, the stable configuration of **5** can be determined by the quantum chemical calculation method.

Answer: Thanks for your kindly suggestion. We have given serious consideration to this comment. The structure of compound **5** was finally elucidated to be a novel enamine-containing PPAP based on the following significant evidence and analysis:

(1) The ^1H - ^{15}N HSQC spectrum of **5** displayed obvious correlations from both the two protons at δ_{H} 12.08 and 6.09 to the nitrogen signal, which demonstrated the presence of a typical primary amine group in **5**.

Expansion of ^1H - ^{15}N HSQC spectrum of compound **5**

(2) In the ^1H - ^{15}N HSQC spectrum of **5**, the chemical shift of the nitrogen signal observed at δ_{N} 126.4 further demonstrated the presence of a C-N single bond (range from 90 to 140 ppm) rather than a C=N double bond in imines (range from 305 to 375 ppm).

Table of ^{15}N chemical shifts (source: Steffen's Chemistry Pages, facts and tables around chemistry, <https://wissen.science-and-fun.de/chemistry/spectroscopy/15n-chemical-shifts/>)

(3) In the ^1H - ^1H COSY and ROESY spectra, there are obvious correlations between the aforementioned two protons.

Expansion of ^1H - ^1H COSY spectrum of compound **5**

Expansion of ROESY spectrum of compound **5**

Therefore, we make sure that compound **5** is correctly identified as an enamine-containing PPAP based on the above analysis and evidence. During the isolation and identification of compound **5**, we found that this compound is relatively stable. However, the NMR spectra of garciyunnanimines A–C showed nearly identical chemical shifts of C-3 (δ_{C} 110.4–111.1) and C-15 (δ_{C} 171.5–172.0) compared to **5**, suggesting that they possess a same primary enamine moiety rather than an imine group. So, it is necessary to revise the structures of garciyunnanimines A–C to their corresponding PPAP enamines. The previous Figure 6 (structural reassignment of

garciyunnanimines A–C) in the article has been relocated to the Supplementary Information as supplementary **Figure S13** at **Page 22**, in order to comply with the journal style and formatting requirements of Communications Chemistry [a maximum of 10 display items (figures and tables) for ~5000 words]. In addition, the stable configuration of **5** has been determined by single-crystal X-ray diffraction and Quantum chemical ECD calculations. Please review the following single-crystal X-ray structure and calculated ECD spectra. Thank you again.

X-ray crystallographic structure of compound **5** (Figure 4, Page 8)

Quantum chemical ECD calculations of compound **5** (Figure 5, Page 11)

III. Comments from Reviewer 3

This manuscript described five nitrogenous polycyclic polyprenylated acylphloroglucinols (PPARs) and their anti-neuroinflammatory activity. Their structures were determined by spectroscopic analysis, ECD and NMR calculations, and X-ray. The structures with α -aminonitrile moiety were interesting. This manuscript could be accepted after the following issues are solved.

1. For the structural elucidation for compound **1**, the evidence for the presence of CN group was not enough. The IR spectrum should have a characterized peak at 2210 cm^{-1} . I carefully checked the IR data of the five compounds in Experiments part, no 2210 cm^{-1} was provided. Please recheck the IR spectrum and rewrite the structural elucidation for compound **1**.

Answer: Firstly, we sincerely appreciate your valuable identification and kind comments. We are sorry for the lack of sufficient evidence in the IR spectra to identify the presence of the nitrile (-CN) group in these new compounds. It is indeed true that a nitrile group gives a medium-intensity band in the triple-bond region of the IR spectrum (2320–2100 cm^{-1}). However, it is difficult to detect this bond in the previous IR spectra because of the low testing amounts of **1–4**. We have increased their testing amounts and re-measured the IR spectra. The new IR spectrum displayed a specific absorption band at 2308 cm^{-1} due to the nitrile group. Therefore, we replaced the previous IR spectra and IR data with these new spectra (**Figures S36, S58, and S72** in the revised Supplementary Information) and new data (**Pages 20 and 21** in the revised Manuscript). Please review them again.

Furthermore, the structural elucidation for compound **1** has been rewritten according to your comments. The revised portions were marked in yellow color at **lines 71–80, Pages 4–5**. Please kindly review this section once again in the revised manuscript. Thanks.

New IR spectrum (film on KBr pellet) of 1/2.

New IR spectrum (film on KBr pellet) of 3/4.

- In Fig.3, some ROESY information are not needed, for example the two hydrogens of NH₂ in compound 5. Please recheck all the ROESY correlations.

Answer: Following your kind suggestion, we have thoroughly rechecked all the ROESY correlations in Fig. 3 based on their respective ROESY spectra. The redundant

ROESY correlations were removed, and the new figure was redesigned in a more concise way (**line121, Page 7**). Please review the revised **Fig .3** below.

Previous Fig. 3

Revised Fig. 3

Finally, I would like to express my sincere gratitude for your valuable comments and suggestions.